# A Scoping Review of Galectin-3 as a Biomarker of Cardiovascular Diseases in Pediatric Populations

**DOI:** 10.3390/ijerph19074349

**Published:** 2022-04-05

**Authors:** Ewa Smereczyńska-Wierzbicka, Radosław Pietrzak, Bożena Werner

**Affiliations:** Department of Pediatric Cardiology and General Pediatrics, Medical University of Warsaw, 02-091 Warsaw, Poland; ewa.wierzbicka@uckwum.pl (E.S.-W.); bozena.werner@wum.edu.pl (B.W.)

**Keywords:** galectin-3, biomarker, heart failure, pediatric cardiology

## Abstract

Galectin-3 as a cardiac biomarker has proven to be a significant prognostic tool in adults. However, it has not yet been established in the pediatric population as a biomarker in daily clinical practice. The aim of the study was to summarize the current knowledge on galectin-3 as a biomarker in children with cardiac conditions by reviewing the literature. Bibliographic databases such as PubMed, Web of Science and Embase were searched, and consequently twelve articles met the inclusion criteria. Supplemental handsearching of references delivered one additional source. These prospective studies concerning galectin-3 as a cardiac biomarker present analyses performed in cohorts composed of healthy children and children with cardiovascular diseases. The results, despite being based on small cohort studies, inform that galectin-3 could serve as a potential biomarker in cardiovascular risk stratification in children with heart failure, arrhythmia, Kawasaki disease or in patients undergoing cardiac surgery. The evidence for the usefulness of galectin-3 in the assessment of such pathologies as idiopathic dilated cardiomyopathy, coarctation of the aorta, functionally univentricular heart or tetralogy of Fallot were not completely confirmed. Galectin-3 seems to be a promising biomarker; however, there is a need for further research to establish its use in daily clinical practice.

## 1. Introduction

### 1.1. Galectin-3—General Information

Galectin-3 protein is a novel biomarker, known mostly as an indicator of inflammation and fibrosis. It is a member of the betagalactosidase-binding galectin protein family and contains a carbohydrate-recognition domain (CRD). Galectin-3 is a chimera-type galectin with a single CRD connected to an N-terminal domain [1]. In the human genome, galectin-3 is encoded by the LGALS3 gene located on chromosome 14, locus q21–q22 [2]. Biological research revealed that galectin-3 can be found in the cellular nucleus, cytoplasm, and cell surface, as well as in the extracellular matrix [3,4,5]. Galectin-3 is produced and secreted by many cells and tissues, and is involved in a variety of processes such as inflammatory reaction, apoptosis, or heart remodeling [5,6,7]. Figure 1 summarizes its possible mechanisms of action in cardiovascular pathologies.

### 1.2. Galectin-3 in Pediatrics

The understanding of the mechanisms of galectin-3 activity gave rise to many studies concerning the protein as a biomarker in clinical practice in adults as well as children. 

In the pediatric population, apart from the cardiovascular diseases discussed below, galectin-3 was studied in a variety of illnesses; one example of which was children with hepatitis B, proving that serum galectin-3 levels may be a beneficial indicator of chronicity of hepatitis B infection in children [8]. Borges et al. [9] demonstrated that the evaluation of galectin-3 is helpful in establishing differential diagnoses among pediatric posterior fossa tumors, based on the fact that galectin-3 is highly expressed in several tumors, including brain neoplasms. Moreover, some authors suggest that the expression of galectin-3 in nephrotic syndrome glomerulopathies in children could be regarded in certain individuals as an indicator of an unfavorable prognosis [10]. Significantly higher levels of galectin-3 were also found in patients with osteosarcoma, and the authors assume that serum galectin-3 could serve as a useful biomarker for the evaluation of osteosarcoma progression [11]. What is more, follow-up case-controlled study results by Ezzat et al. [12], suggest that serum galectin-3 levels can indicate activity, severity and progression of juvenile idiopathic arthritis in the pediatric population. Furthermore, the prognostic value of serum galectin-3 in children with sickle cell diseases was assessed in two studies suggesting that galectin-3 can be indicative of myocardial ischemia during vaso-occlusive crisis and a susceptibility factor for vaso-occlusive crisis frequency [13,14].

### 1.3. Galectin-3 in Pediatric Cardiology

Although other biomarkers such as troponin or NT-proBNP are commonly used in pediatric cardiology, their value is limited by some disadvantages. Among these limitations, the alternating, age-dependent and broad normal range, as well as their relatively difficult interpretation, are the major ones [15,16]. As a result, the need for a new, cheap, reproducible, and easily interpreted biomarker still exists. In light of the data indicating that galectin-3 does not correlate with age in healthy children [17,18], it is becoming a new, promising tool in the diagnosis of such cardiovascular system disorders as a heart failure, cardiomyopathies, inflammatory diseases, or congenital heart diseases. 

## 2. The Aim

The aim of the study was to review the literature on galectin-3 as a cardiac biomarker in pediatric populations.

## 3. Methods

The authors performed a comprehensive search according to the Preferred Reporting Items for Systematic Reviews and Meta—Analyses Extension for Scoping Reviews (PRISMA-ScR)—Figure 2. An overview of the terms “galectin-3” AND “cardiology” OR “cardiovascular system” in databases such as PubMed, Web of Science and Embase was conducted. All types of studies meeting the following inclusion criteria were searched: articles in Polish and English, population 0–18 years old. The following exclusion criteria were applied: systematic reviews, metanalyses, case studies, articles published before 2014, research in animals. After excluding duplicates, 20 abstracts were assessed for eligibility. Subsequently, 16 full-text articles were studied, out of which 12 were selected for qualitative analysis. Supplemental handsearching of references provided one additional source. The selected studies were analyzed independently by all researchers. We resolved disagreements on study selection and data extraction by discussion if needed. We selected all the publications concerning galectin-3 in cardiovascular diseases. We excluded articles from the same centers with similar numbers of patients, and also articles with inconsistency in the numerical data. All the studies included in the review were prospective studies (Appendix A).

## 4. Results

Studies assessing galectin-3 as a cardiac biomarker in children with cardiovascular diseases included in this review are listed in Table 1 As a final result, 13 articles published between 2014 and 2021 were taken into consideration. 

### 4.1. Heart Failure

In the studies conducted in pediatric patients with congenital heart diseases, galectin-3 plasma concentration was significantly higher in children presenting with heart failure signs and symptoms than in those without [19,20,21,22]. Moreover, galectin-3 levels correlated positively with the severity of circulatory system insufficiency expressed using the Ross Heart Failure Scale, and was lower in children treated with oral spironolactone than in those without [19,21]. The evidence of correlation between galectin-3 plasma concentration and chosen echocardiographic parameters of left ventricular function, both systolic and diastolic, exists in the literature and was presented in the studies of Kotby et al. and Saleh et al. [19,21]. In those studies, significant positive correlations between serum galectin-3 levels and left ventricular echocardiographic parameters—such as diastolic and systolic diameter, end-diastolic volume—as well as pulsed wave and tissue doppler measurements—such as mitral E and A wave velocity and myocardial E^I^ and A^I^ velocity—were found. Concurrently, significant negative correlations of galectin-3 plasma concentrations in relation to contractile function parameters such as ejection fraction and shortening fraction were presented.

In a trial by Woulfe et al. [23], the expression of galectin-3 in myocardial biopsy was analyzed. It was increased in the left ventricular myocardial tissue in adults, but this finding was not confirmed in pediatric patients, regardless of the occurrence of fibrosis in this group. The authors concluded that this was a consequence of the lower intensity of fibrosis found in children with idiopathic dilated cardiomyopathy in comparison to adult tissue. 

### 4.2. Kawasaki Disease

In children in the pre-treatment phase of Kawasaki disease, the plasma concentration of the galectin-3 was found to be significantly higher than in the control group [24]. However, the galectin-3 level was not significantly different in children with and without aneurysms in the acute phase of the disease. A relationship between galectin-3 plasma concentration and inflammatory state exponents was not found either. Nevertheless, persistent, elevated galectin-3 levels were still detected in the late convalescent phase of the disease (more than a year) and its plasma concentration was higher in children with aneurysms than in those without coronary artery involvement. Additionally, the autopsies of the myocardium or coronary artery aneurysms in patients who died or underwent transplantation over the course of Kawasaki disease revealed galectin-3 expression cells. Interestingly, heterogeneity in the galectin-3 expression pathway was found in the examined tissue—two different patterns were present in the acute phase and in the late convalescent phase of the disease where inflammatory or fibrotic myocardial cells, respectively, were responsible for galectin-3 expression.

### 4.3. Congenital Heart Diseases Treated Surgically

Galectin-3 is a strong predictor of poor prognosis in the early follow up after surgery in children treated for such congenital heart diseases as a ventricular septal defect, atrioventricular septal defect or pulmonary valve regurgitation after tetralogy of Fallot repair [25]. Preoperatively, an elevated galectin-3 level was predictor of 365-day readmission or mortality in this cohort. Moreover, an unfavorable course of post-operative intensive care with acute kidney injury in children ≥ 2 years old can be predicted by high galectin-3 plasma concentrations, as was proved in a large, multicenter study by Greenberg et al. [26]. However, the data is inconsistent, as in a smaller study by Parsons et al. [27], pre- and post-operative plasma concentrations of galectin-3 in 162 children undergoing cardiac surgery did not significantly differ between patients with and without acute kidney injury. On the other hand, the data regarding aortic coarctation and tetralogy of Fallot are less promising. Firstly, according to the trial of Frank et al. [28], galectin-3 is not a useful biomarker for persistent left ventricular remodeling prediction during one-year follow-up in children undergoing surgical repair of aortic coarctation. Secondly, a prospective study in 16 patients with a broad range of age after surgical repair of tetralogy of Fallot [29] did not reveal any correlation between galectin-3 plasma concentrations and right heart invasive hemodynamic measurements such as right ventricular end-diastolic pressure and mean right atrial pressure in patients undergoing catheterization for pulmonary valve replacement. Moreover, significant correlation was not found in relation to cardiac magnetic resonance data, such as indexed right ventricular and left ventricular end-diastolic volumes and ejection fraction. Moreover, strong evidence for its poor utility as a risk stratification biomarker in young patients with congenital heart disease treated surgically comes from a study by Bosch et al. [30]. In children with Fontan circulation, galectin-3 showed no relation to cardiac function nor long-term postoperative outcome in this cohort. The authors performed a broad assessment of 133 patients with the evaluation of the incidence of adverse cardiac events (cardiac death, out of hospital cardiac arrest, heart transplantation, cardiac reintervention, hospitalization, and cardioversion/ablation for arrhythmias) and cardiovascular status in stress cardiac magnetic resonance imaging with dobutamine and cardiopulmonary exercise testing, revealing no statistically significant correlation to galectin-3 plasma concentrations.

### 4.4. Arrhythmia

In the study of Pietrzak et al. [31], galectin-3 was found as a promising factor in the assessment of the clinical significance of ventricular heart rhythm disturbances in adolescents. The authors found that the plasma concentration of galectin-3 in patients diagnosed with ventricular arrhythmia was significantly higher compared to its level in healthy children. Moreover, the research demonstrated a statistically significant moderate positive correlation between galectin-3 plasma concentrations and left ventricular diastolic diameter, as well as a moderate negative correlation between its plasma concentration and left ventricular ejection fraction. The authors concluded that galectin-3 can be a promising tool in the evaluation of adolescents with ventricular heart rhythm disturbances who are at risk of structural cardiovascular system pathology, but further investigation is necessary to establish its value in the clinical setting. 

## 5. Discussion

Management of patients with cardiovascular diseases usually requires risk stratification for complications and further clinical course of the disease. Therefore, reliable prognostic biomarkers that help to improve the accuracy of the diagnosis and the way of the treatment are still desirable. The understanding of the mechanisms of galectin-3 activity gave rise to the consideration of this biomarker as a valuable tool in clinical practice, either in the initial phase of the disease or during follow up in various circulatory system pathologies, including those seen in children. For these reasons, we reviewed the literature on galectin-3 as a cardiac biomarker in the pediatric population.

The published studies in children revealed that galectin-3 is elevated in diseases which are connected with volume overload and is thus related to the clinical signs. The results clearly presented the differences of the concentrations of this biomarker in children presenting varying degrees of heart failure symptoms—from the asymptomatic to patients presenting symptoms at rest—as correlated by the Ross Heart Failure Classes [19,21]. Moreover, its relationship with the hemodynamical parameters is clear. In the studies of Mohammed et al., Kotby et al. and Saleh et al. [19,20,21], it was correlated with the systolic function of the left ventricle expressed as the calculation of ejection fraction and fractional shortening in percentages, as well as with left ventricular diameter. These data may be explained by the nature of the biomarker. Galectin-3 expression increases the inflammation or degeneration of the cells due to the stretch or squeeze of the myocardial fibers in the initial stages of the disease, thus being the marker of the pathological pathway leading from inflammation to fibrosis and the inevitable remodeling of the heart muscle [32]. As a marker of fibrosis, galectin-3 also expresses the effectiveness of the cardioprotective nature of spironolactone in children with chronic heart failure due to congenital heart diseases in whom anti-aldosterone therapy is conducted broadly as a prophylaxis against pathological remodeling of the myocardium [19].

In summation, galectin-3 could be a valuable prognostic biomarker in the assessment of heart failure in children; however, the data are still sparse and further trials are necessary to evaluate the value of the biomarker and its relationship to hemodynamical dysfunction. 

The role of galectin-3 is widely investigated in adults with heart failure. The trials show galectin-3 as a biomarker of prognostic importance, being a significant predictor of mortality and rehospitalization in the population with acute, as well as chronic, heart failure [33,34]. As studied in adults, galectin-3 may be useful for predicting the future development of new onset of heart failure and for follow up not only as a single factor, but mainly in combination with other proteins. Watson et al. [35] first evaluated the value of baseline and interval change values of biomarkers connected with fibroinflammation, such as BNP, hsTroponin-I, interleukin-6, sST2, and galectin-3, to identify patients at risk of heart failure. They investigated the usefulness of these proteins to stratify the prognosis in those patients. They provided evidence for the utility of combining analysis of BNP, hsTroponin-I, and galectin-3 in the prediction of future heart failure with preserved ejection fraction. However, inflammatory state exponents such as interleukin 6 and aST2 were not useful in predicting new onset or for follow up in such patients. Adding galectin-3 to the panel of biomarkers assessed in adult patients with heart failure seems to be of great value in predicting the risk of mortality in this cohort [36]. Thus, the Guidelines for the Management of Heart Failure by the American Heart Association indicate galectin-3 as an additional biomarker for the prognosis and risk stratification in patients with heart failure (COR IIb) [37]. Conversely, the European Society of Cardiology in Heart Failure Guidelines emphasize the fact that the current knowledge is not enough to introduce galectin-3 into clinical practice [38].

Galectin-3 may also be a valuable tool to establish the extent of tissue destruction in chronic inflammation. In Kawasaki disease, galectin-3 plasma concentration is higher in children in whom chronic inflammation drives aneurysmal changes in the coronary arteries, as compared to those patients in whom the disease takes the form of the progressive condition. 

To some extent, this hypothesis is confirmed by the data showing that the results of surgical treatment of congenital heart disease are worse in patients with increased levels of galectin-3 than in those with normal plasma concentrations of the protein [25]. These results raise the question of whether the increase of galectin-3 may precede irreversible fibrosis, or if it only reflects the constant injury of the tissue. There are some data in the literature regarding adult patients with arterial hypertension [39], indicating that it is a useful biomarker in the initial stages of the left ventricular myocardial overload, and therefore may be used for the detection of early cardiac remodeling. The assumption that galectin-3 is an indicator of early injury can also be supported by the aforementioned decrease of galectin-3 plasma concentrations after the use of spironolactone in children with volume overload. 

The prognostic nature of galectin-3 was also a point of research for Greenber et al. and Parsons et al. [26,27]. In relation to the postoperative condition of patients with congenital heart diseases after cardiosurgical treatment, galectin-3 was studied in order to predict the risk of acute kidney injury during post-operative care, and the results of the larger cohorts are promising. Complications, out of which acute kidney injury is of great importance, are always the factors of poor prognosis. Therefore, the screening of the patients for increased concentration of galectin-3, in which we could expect said complications, would be of significant value and would lead to the improvement of the quality of care.

Contrarily, Zegelbone et al. [29], in his study of patients after the surgical repair of tetralogy of Fallot, did not find any correlation between galectin-3 plasma concentration and right ventricular invasive hemodynamical parameters, nor measurements derived by cardiac magnetic resonance. Although the study was conducted on a small sample size (only 16 patients), these results raise doubt of whether galectin-3 may be a useful marker of myocardial damage of the right ventricle in patients after correction of congenital heart diseases. Nevertheless, plasma level of galectin-3 reflects the extent of fibrosis or damage rather than hemodynamical disturbances. 

Although Karatolios et al. [40] reported that galectin-3 activity assessment in myocardial tissue is useful in predicting cardiac remodeling in adults suffering from idiopathic dilated cardiomyopathy, this finding in pediatric patients was not confirmed by Woulfe et al. [23]. The authors hypothesized that the low expression of galectin-3 relates to the shorter time span following the diagnosis in pediatric patients in comparison to adults (respectively 1 year vs. 5 years up to the heart transplant). Furthermore, they suggested that the rapid progression of the disease in children comes from an inappropriate/different inflammatory response to the “injuring factor” in the clinical course of the disease. Thus, galectin-3 levels, as an inflammatory state exponent, are not as increased in children as it is in adults.

The results presented in the study of Frank et al. [28] were not promising, either. They found that galectin-3 did not predict persistent left ventricular hypertrophy nor abnormal wall thickness in children after the cardiosurgical treatment of aortic coarctation. However, that trial was also conducted in a small cohort (27 participants) with a broad age range (2 months—12 years) so further research is needed, and the hypothesis remains to be clarified (Appendix A).

Both children and adults with functionally univentricular hearts with Fontan circulation were assessed by researchers in order to determine the potential of galectin-3 as a prognostic biomarker in this cohort. A large group of 133 young patients at a median age of 13.2 years was studied, and no evidence for distinguishing high-risk patients (concerning higher incidence of adverse events such as arrhythmia, reintervention, death, cardiac arrest or heart transplantation) was found [30]. The authors found more optimistic results for predicting undesirable outcomes, such as nonelective hospitalization or death adult patients (median age 30.5 years) with single-ventricle Fontan circulation [41]. What needs to be stressed is that, in this specific cohort, the consequence of the disease is progressive multiorgan failure and fibrosis, which can be the reason for the correlation of this marker and adverse events in older patients over time. As it was shown in those studies, the analysis of galectin-3 in adults could serve as a clinically valuable tool, which is contrary to the results in children over the median follow-up time of 9.7 years after the Fontan procedure where no such correlation was found.

Galectin-3 was also broadly studied in grown-ups with congenital heart diseases, and the results could serve as a direction for future research in younger cohorts. In the study by Frogoudaki et al. [42] of galectin-3 as a risk factor for major cardiovascular events defined as death, hospitalization, worsening functional class or cardiac intervention did not produce optimistic results; however, the concentration of the marker correlated with arrhythmias and was significantly higher in the case of patients presenting supraventricular or ventricular tachycardia. Moreover, the authors found a correlation between galectin-3 and global longitudinal strain in adults with congenital heart diseases. Galectin-3, as a marker of ventricular function, was also assessed in a cohort of adults with a systemic right ventricle by Geenen et al. [43] and only weak correlations were found in relation to right ventricle global longitudinal strain; however, again, higher galectin-3 levels were associated with an increased risk of arrhythmias. An interesting study in adult patients with arrhythmogenic right ventricular cardiomyopathy by Oz et al. [44] and implanted defibrillators adds to the evidence for galectin-3 being useful in the risk stratification of arrhythmias. Emphasizing fibrosis as the source of the disease, the authors found galectin-3 to be predictive of ventricular tachycardia and ventricular fibrillation in this cohort. An interesting report by Geenen et al. [45] assumed that we have a significant decrease of galectin-3 one day post-atrial septal defect closure in adults, suggesting that it could reflect shunt cessation and subsequently volume overload of the right ventricle. Moreover, in 2020, the results delivered by Kowalik et al. [46] showed galectin-3 to be useful in patients with an overloaded right ventricle; however, only in those with congenitally corrected transposition of the great arteries. No significant correlation was found in adults with Eisenmenger syndrome.

Considering the results in adults with congenital heart disease, there is an optimistic view for galectin-3 to serve as a valuable marker, especially for the prediction of life-threatening arrhythmias, and hopefully for ventricular function assessment and risk stratification in specific groups of patients. 

In the literature, many trials exist in the adult population with clinical problems such as pulmonary hypertension, hypertrophic cardiomyopathy or arrhythmogenic right ventricular cardiomyopathy, showing that galectin-3 plasma concentrations are higher in those patients when compared to healthy individuals. Galectin-3 levels reflect heart remodeling. This was proven by the relationship of the biomarker level and echocardiographic measurements. In the study performed by Fenster et al. [47] in patients with pulmonary hypertension, the authors detected that galectin-3 was associated with right ventricular echocardiographic and cardiac magnetic resonance indices, such as ejection fraction, systolic and diastolic volumes, left ventricular mass index or systolic pressure and strain. On this basis, the authors of *Expert consensus statement on the diagnosis and treatment of pediatric pulmonary hypertension* emphasize the fact that galectin-3 is a promising biomarker in adult pulmonary arterial hypertension, setting the direction for further research [48]. 

The direction of research in children leads, as it does in adults, to the field of arrhythmias, where galectin-3 is researched for its ability to detect patients with a worse course of the disease and assumingly with poorer prognosis. The main area of focus is atrial fibrillation. Galectin-3 plasma concentrations predict arrhythmia recurrence following a single ablation procedure, whilst the data considering the value of galectin-3 in the anticipation of the effectiveness of the ablation are conflicting. Furthermore, galectin-3 was independently associated with atrial remodeling in patients with chronic atrial fibrillation [49,50,51]. Gurses et al. [50] questioned galectin-3 in patients with atrial fibrillation and preserved left ventricular function, assuming that the significantly elevated biomarker was correlated with left atrial volume index in this population. Coherent results were found in a study by Wu et al. [52], showing higher plasma galectin-3 levels in patients with persistent atrial fibrillation and potency of galectin-3 as a predictor of atrial fibrillation recurrence after catheter ablation, confirmed by Clementy et al. [53] in a larger cohort. Interestingly, a higher recurrence rate of atrial fibrillation was confirmed in patients undergoing surgical ablation and having higher galectin-3 levels after the procedure [54]. There are some data in the literature showing that it is helpful in predicting ventricular arrhythmias in patients with dilated cardiomyopathy [55]. In light of these data, some promising results in this area were reported in adolescents with ventricular arrhythmia. The study of 25 patients revealed that galectin-3 is elevated and corresponds with left ventricular size and function preservation assessed in echocardiography, as well as in cardiac magnetic resonance [31]. However, those are the initial reports in a small cohort and are not conclusive enough to introduce galectin-3 into the standard panel of measurements in patients with ventricular arrhythmia in daily practice. 

Last but not least, it is worth mentioning the report of Dencker et al. [56] of a healthy cohort of 170 children. The aim of the study was to assess the relationship between galectin-3 levels and total body fat, abdominal fat, body fat distribution, aerobic fitness, blood pressure, left ventricular mass, left atrial size, and increase in body fat over a two-year period in a population-based sample of children. The interesting results showing higher galectin-3 levels in those with more body fat and a more abdominal distribution of fat or with a greater left ventricular mass and increased left atrial size led to the conclusion that, in children, galectin-3 could serve as a potential risk factor for the development of cardiovascular diseases in adulthood. Highlighting the growing problem of obesity in children which leads to cardiological system dysfunction later in life, the direction for further research seems clear.

## 6. Conclusions

As the first studies of galectin-3 in children with cardiovascular diseases are very optimistic, there is need for further research. We can assume that galectin-3 in pediatric cardiology appears to be a potential biomarker; however, there is still insufficient data to be able to introduce galectin-3 in clinical practice guidelines. Further investigations should doubtlessly focus on multicenter studies consisting of larger cohorts. A potential and promising direction for research is galectin-3 concentrations in children with congenital heart diseases, heart failure symptoms, arrhythmias or myocarditis (Figure 3).

### Study Limitations

The limitations of the study include the lack of statistical meta-analysis, as well as heterogeneous data and small sample sizes (Appendix A).

## Figures and Tables

**Figure 1 ijerph-19-04349-f001:**
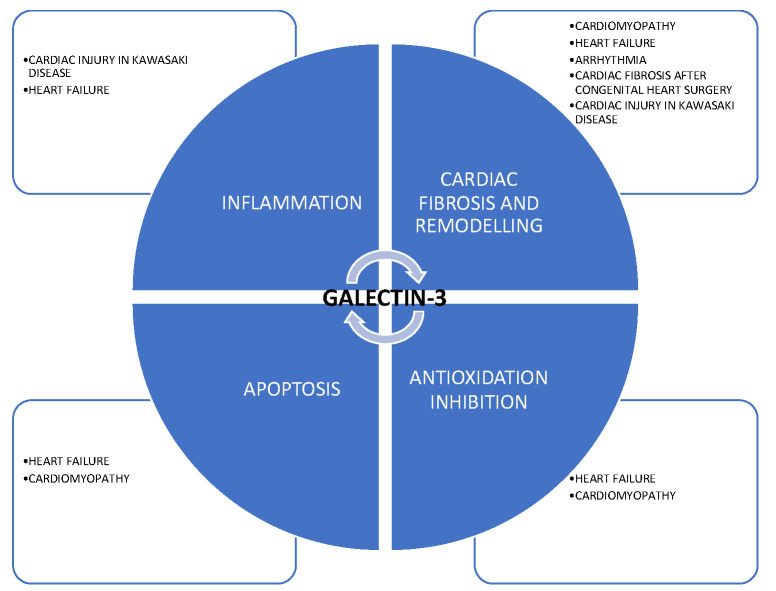
Possible mechanisms of action of galectin-3 in cardiovascular pathologies.

**Figure 2 ijerph-19-04349-f002:**
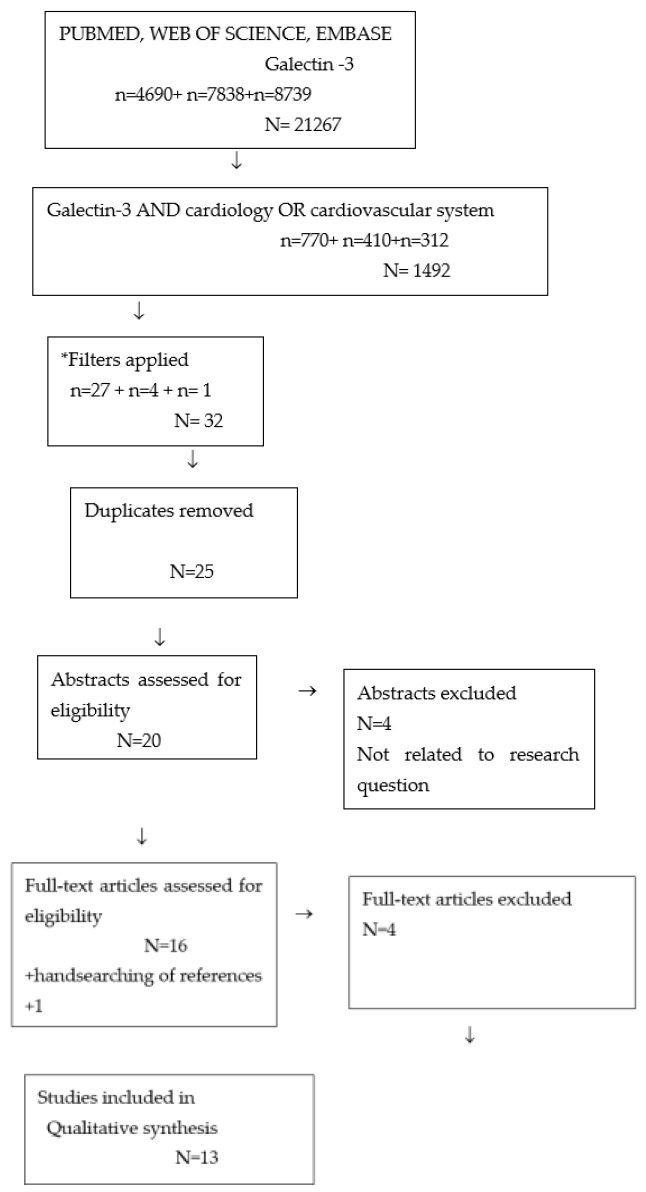
Search strategy. * Filters applied: Child: birth–18 years; Newborn: birth–1 month; Infant: birth–23 months; Infant: 1–23 months; Preschool Child: 2–5 years; Child: 6–12 years; Adolescent: 13–18 years; Publication date: 1 January 2014–24 September 2021; Humans. n = results in each database. N = total results.

**Figure 3 ijerph-19-04349-f003:**
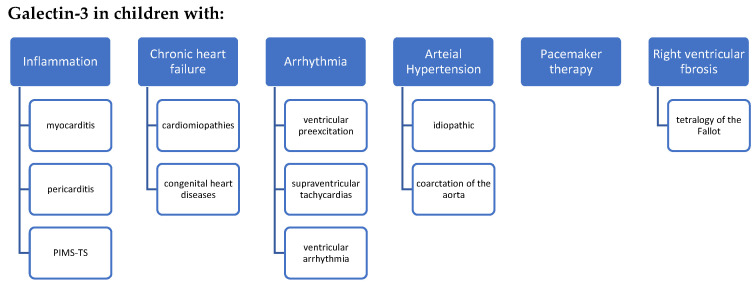
Future research directions for galectin-3 in pediatric cardiology. PIMS-TS: pediatric inflammatory multisystem syndrome temporally associated with SARS-CoV-2.

**Table 1 ijerph-19-04349-t001:** Galectin-3 in pediatric populations with cardiovascular disease.

Reference Publication Year/[Reference Number]	Aim of the Study	Results
Mohammed et al., 2014/[20]	G3 for HF prediction in children with CHD	G3 higher in HF vs. non-HF (*p* < 0.001) and controls (*p* < 0.001). Positive correlation of G3 and: ○ LA diameter (r = 0.264, *p* = 0.001), ○ LV diameter r = 0.364, *p* = 0.004), ○ PAP (r = 0.452, *p* < 0.05). Negative correlation of G3 and: ○ FS% (r = −0.309, *p* = 0.016), ○ EF% (r = −0.314, *p* = 0.014).
Numano et al., 2015/[24]	G3 in Kawasaki disease (acute and convalescent patients) G3 stained in the myocardium and coronary arterial walls autopsy and cardiac transplant cases	G3 higher in adults after Kawasaki disease with giant aneurysms (*p* < 0.05). G3 higher in children with Kawasaki disease (*p* < 0.05) compared to healthy controls irrespective of disease phase (acute, early convalescent and late convalescent group) and coronary artery status (presence of coronary artery aneurysms). G3 higher in children with coronary artery aneurysms compared to G3 in patients with no coronary artery aneurysms (*p* < 0.05) in the late convalescent phase group. Autopsy/explanted heart tissues—two distinct patterns of G3 expression: ○ acute phase—G3 expressed by infiltrating inflammatory cells; ○ the late convalescent phase in patients with giant aneurysms—G3 expressed by spindle shaped cells in the densely fibrotic regions of myocardium and arterial media.
Kotby et al., 2016/[19]	G3 in children with chronic HF G3 correlation to disease severity and progression	G3 increased in HF compared to control group (*p* < 0.001). G3 elevated in patients with EF > 50% compared to patients EF < 50% (*p* = 0.194). G3 and Ross HF classes positive correlation (r = 0.73, *p* < 0.001). G3 elevated in patients without spironolactone compared to G3 in patients on spironolactone (*p* = 0.049). G3 and echocardiographic parameters positive correlation: ○ LVEDD (*p* = 0.049), ○ LVESD (*p* = 0.04), ○ EDV (*p* = 0.024), ○ LVMI (*p* = 0.001), ○ RVESP (*p* < 0.001), ○ E wave (*p* < 0.001), A wave (*p* = 0.001), ○ E/A ratio (*p* = 0.01), ○ Em (*p* = 0.02), Am (*p* = 0.01), Em/Am (*p* = 0.04), ○ E/Em (*p* = 0.01). G3 and echocardiographic parameters negative correlation: ○ FS% (*p* = 0.028), ○ EF% (*p* = 0.024), ○ Sm (*p* = 0.01).
Woulfe et al., 2017/[23]	Age-related differences in pathologic fibrosis and selected fibrosis gene expression (i.a. noncoding G3) in children and adults undergoing transplantation owing to end-stage IDC	The expression of noncoding G3 in IDC LV tissue elevated in adults but not in the pediatric population compared with age-matched control samples. Noncoding G3 expression not changed with the pathologic presence of fibrosis in pediatric IDC patients.
Frank et al., 2018/[28]	Association between G3 and echocardiographic persistent LV abnormalities at intermediate-term follow-up in patients with CoA undergoing surgical repair	G3 higher in the neonatal population than in older children (*p* = 0.02) at the pre-op time point. No linear relationship of G3 and LVMI or RWT in post-op nor in the follow-up (1-year post-op) period. G3 unchanged from pre-op to post-op period.
Zegelbone et al., 2019/[29]	Association of G3 and right heart volume/pressure overload from pulmonary valve insufficiency and/or stenosis before pulmonary valve replacement	No significant correlations between G3 and right heart hemodynamic measurements, both invasive and MRI derived.
Elhewala et al., 2020/[22]	G3 in children with CHD	G3 higher in children with CHD compared to control group. G3 and Ross HF scale positive correlation. G3 higher in children with HF symptoms compared to those without HF.
Parker et al., 2020/[25]	G3 association of 365-day readmission or mortality after paediatric congenital heart surgery	Preoperative G3 as a strong and significant predictor of readmission or mortality (significant association of unadjusted preoperative log-transformed galectin-3 and a two-fold increase in risk of 365-day readmission or mortality).
Saleh et al., 2020/[21]	G3 in children with HF secondary to CHD and its correlation with mortality in this group	G3 increased in CHD patients with HF compared to control group (*p* < 0.001). G3 higher in children with CHD and HF compared to children with CHD and no HF symptoms (*p* < 0.001). G3 and Ross HF classes positive correlation (r = 0.68 *p* < 0.001). No significant correlation between G3 and HF mortality. G3 and echocardiographic parameters positive correlation: ○ LVESD (*p* < 0.003). G3 and echocardiographic parameters negative correlation: ○ EF% (*p* < 0.001), ○ FS% (*p* < 0.001).
Parsons et al., 2020/[27]	G3 for risk stratification of AKI in children with CHD undergoing cardiac surgery	Pre- and post-operative G3 not associated with AKI.
Greenberg et al., 2021/[26]	G3 for risk stratification of post-operative AKI in children with CHD after cardiac surgery	Pre- and post-operative G3 associated with AKI in children ≥ 2 years old.
Pietrzak et al., 2021/[31]	G3 impact on myocardial tissue preservation in adolescents with ventricular arrhythmia	G3 higher in children with ventricular arrhythmia compared to control group (*p* < 0.001). G3 not differentiated in children with or without complex arrhythmia.
Bosch et al., 2021/[30]	G3 association of cardiac function and adverse outcome in a young Fontan cohort	No relation of G3 and cardiac function nor long-term outcome. No potential of G3 in risk stratification of patients who have undergone the Fontan procedure.

Legend: G3—galectin-3 plasma concentration; HF—heart failure; CHD—congenital heart diseases; LA—left atrium; LV—left ventricle; PAP—pulmonary artery pressure; FS—fractional shortening in %; EF—ejection fraction in %; LVEDD—left ventricular end-diastolic diameter; LVESD—left ventricular end-systolic diameter; EDV—end-diastolic volume; LVMI—left ventricular mass index; RVESP—right ventricular end-systolic pressure; E wave—early diastolic mitral velocity; A wave—late diastolic mitral velocity; Em—peak early diastolic mitral annulus velocity; Am—peak late diastolic mitral annulus velocity; Sm—peak systolic mitral annulus velocity; IDC—idiopathic dilated cardiomyopathy; CoA—coarctation of the aorta; RWT—relative wall thickness; MRI—magnetic resonance imaging; AKI—acute kidney injury.

## Data Availability

Data is contained within the article.

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
