# Peer review of "A Scoping Review of Galectin-3 as a Biomarker of Cardiovascular Diseases in Pediatric Populations"

_ijerph, 2022, doi:10.3390/ijerph19074349_

Round 1

Reviewer 1 Report

The authors of the submitted manuscript undertook a systematic summarization of the galectin-3 usages in multiple pediatric cardiovascular diseases. The manuscript is an interesting summarization of the current knowledge of galectin-3, which is of great value in science per se. But this paper has several major and minor concerns.

  1. This review shared a huge similarity with a meta-analysis report, which can confuse the readers.
  2. This review set up inclusion and exclusion criteria for the enrolled publications, making the paper offer less information as it could. Suggestion is to reshape this manuscript as a review paper by deleting the method part and adding more published information.
  3. Galectin-3 pathway was found to be related to many other clinical factors. It upregulates interleukin-6, another biomarker of cardiovascular disease. And galectin-3 also plays a regulatory role in immunometabolism and fibrogenesis, shown by the clinical biomarker Interleukin-33. For the benefit of the readers, it is suggested to discuss more about the relationship between galectin-3 and other available biomarkers like above.

Author Response

Reviewer 1

  1. The authors of the submitted manuscript undertook a systematic summarization of the galectin-3 usages in multiple pediatric cardiovascular diseases. The manuscript is an interesting summarization of the current knowledge of galectin-3, which is of great value in science per se. But this paper has several major and minor concerns.

Response: Thank you for this comment.

  1. This review shared a huge similarity with a meta-analysis report, which can confuse the readers.

Response: Thank you for this suggestion, however we decided to perform our trial according to scoping review protocol (PRIMSA – ScR), which is the most common and, in our opinion, the most valuable form of the review articles.  All the similarities between this form of review trial and meta-analysis comes from the fact that we need to meet all the criteria for scoping review protocol.  We believe that after we mentioned in the title as this is a scoping review the reader will find it clear. Moreover, we pointed out in the main text that we followed the protocol for the scoping review.

  1. This review set up inclusion and exclusion criteria for the enrolled publications, making the paper offer less information as it could. Suggestion is to reshape this manuscript as a review paper by deleting the method part and adding more published information.

Response: As we mentioned in the first comment, we followed the scoping review protocol (PRIMSA – ScR) and it is obligatory to include method part. What is more, the other reviewers asked for more details in method part.

  1. Galectin-3 pathway was found to be related to many other clinical factors. It upregulates interleukin-6, another biomarker of cardiovascular disease. And galectin-3 also plays a regulatory role in immunometabolism and fibrogenesis, shown by the clinical biomarker Interleukin-33. For the benefit of the readers, it is suggested to discuss more about the relationship between galectin-3 and other available biomarkers like above.

Response: Thank you for pointing this out. We certainly agree therefore we enriched our review with this topic.

Reviewer 2 Report

The topic is of interest, however major revisions would be required before considering for publication. 

The authors need to revise the table which currently contains an overwhelming amount of information and in revision need to focus more on the possible perspectives of  galectin-3 application  in children - could the authors present the potential application i.e. in the form of figure/visual graph ? 

Are there any differences between the Gal-3 concentration changes in children and among the adults? Since Gal-3 is not dependent on age in healthy children (as reported by the authors), can paediatrician learn something from the studies in the adults? this reviewer believes that the manuscript would benefit from such a paragraph and a table (as a supplementary table) addressing the knowledge on gal -3 already obtained in adults vs children - focusing on the corresponding areas of research (congenital heart diseases, heart failure symptoms, arrhythmia or myocarditis.)

Author Response

Reviewer 2

1.The topic is of interest, however major revisions would be required before considering for publication. 

Response: Thank you for this comment, we revised the manuscript concerning all the comments.

  1. The authors need to revise the table which currently contains an overwhelming amount of information and in revision need to focus more on the possible perspectives of  galectin-3 application  in children - could the authors present the potential application i.e. in the form of figure/visual graph ? 

Response: Thank you for pointing this out. We revised the table content, and we believe it is clearer. We prepared the table with the possible perspectives of research of galectin-3 and its application in children with cardiovascular diseases. Hope you find it valuable.

  1. Are there any differences between the Gal-3 concentration changes in children and among the adults? Since Gal-3 is not dependent on age in healthy children (as reported by the authors), can paediatrician learn something from the studies in the adults? this reviewer believes that the manuscript would benefit from such a paragraph and a table (as a supplementary table) addressing the knowledge on gal -3 already obtained in adults vs children - focusing on the corresponding areas of research (congenital heart diseases, heart failure symptoms, arrhythmia or myocarditis.)

Response: Thank you very much for pointing this out.However, since many reviews have already been carried out on adults, repeating the same information would not improve the scientific value of our study  and exceeds the purpose  of our trial. Moreover, after we combine the review in adults and children and want to describe the topic precisely the size of the article would exceed the demands of the editor’s rules.

Reviewer 3 Report

In general, it is a well written manuscript about the role of galectin-3 in cardiovascular disease in pediatric patients. The evaluation of a single biomarker is a promising strategy that could facilitate diagnosis and prognosis. 

As a comment, I would suggest that the authors specify the number of researchers that took part in the evaluation of the articles, mention if there were any disagreement/conflicts and how they were resolved. Furthermore, you could present in more details the inclusion criteria of the studies in the methods. Also, you could mention this information in the flow-chart for the selected studies. Besides, you should present in the table the type of the study design for the studies that you used in a separate column. 

Author Response

Reviewer 3

  1. In general, it is a well written manuscript about the role of galectin-3 in cardiovascular disease in pediatric patients. The evaluation of a single biomarker is a promising strategy that could facilitate diagnosis and prognosis. 

Response: Thank you very much for this comment.

  1. As a comment, I would suggest that the authors specify the number of researchers that took part in the evaluation of the articles, mention if there were any disagreement/conflicts and how they were resolved. Furthermore, you could present in more details the inclusion criteria of the studies in the methods. Also, you could mention this information in the flow-chart for the selected studies. Besides, you should present in the table the type of the study design for the studies that you used in a separate column. 

Response: Thank you for raising an important issue . We revised the manuscript and corrected according to your suggestions.

Reviewer 4 Report

In this scoping review authors synthesized the research on galectin-3 scope as a biomarker in pediatric patients with cardiovascular diseases (CVDs). Although authors nicely summarized the literature available on the topic but weren’t greatly successful in identifying the gaps and proposing the potential future research that can help validate galectin-3 as a biomarker in pediatric patients with CVDs. This needs to be improved heavily. Please see below

  1. Authors in their conclusive statement went on very broadly to propose what needs to be improved to propose galectin-3 as a biomarker. However, they specifically need to address the scope of clinical research various CVDs (particularly the diseases greatly driven by inflammation and fibrosis).
  2. The figure 1 and 2 seems to be very broad and gives no information as such. Authors can combine figure1 and 2 and can make it more informative by adding the different outcomes that were known to cause by galectin-3 in various CVDs via major signaling pathways that can be targeted pharmacologically. For example, how galectin 3 increase is known to cause inflammation in a particular CVD and that drives disease progression.
  3. There is no information on how the screening process was done by independent reviewers in methodology.
  4. The aim says ‘pediatric population’. Shouldn’t it be the pediatric ‘patient’ population?

Author Response

Reviewer 4

  1. In this scoping review authors synthesized the research on galectin-3 scope as a biomarker in pediatric patients with cardiovascular diseases (CVDs). Although authors nicely summarized the literature available on the topic but weren’t greatly successful in identifying the gaps and proposing the potential future research that can help validate galectin-3 as a biomarker in pediatric patients with CVDs. This needs to be improved heavily. Please see below

Response: Thank you for this comment. We revised the manuscript and took all your comments into consideration.

  1. Authors in their conclusive statement went on very broadly to propose what needs to be improved to propose galectin-3 as a biomarker. However, they specifically need to address the scope of clinical research various CVDs (particularly the diseases greatly driven by inflammation and fibrosis).

Response: Thank you for this suggestion, however we decided to perform our trial according to scoping review protocol (PRIMSA – ScR), which is the most common and, in our opinion, the most valuable form of the review articles.  Following the rules for this protocol our biased opinion has limited impact on the citated articles. Nevertheless, to meet reviewers demands we added a figure  summarizing our proposal for future scope of research including various cardiovascular diseases.

  1. The figure 1 and 2 seems to be very broad and gives no information as such. Authors can combine figure1 and 2 and can make it more informative by adding the different outcomes that were known to cause by galectin-3 in various CVDs via major signaling pathways that can be targeted pharmacologically. For example, how galectin 3 increase is known to cause inflammation in a particular CVD and that drives disease progression.

Response: Thank you for pointing this out. As we agree those figures present general information, we combined those in one figure. However, we believe adding all the information you asked to the figure would make it confusing for the reader.

  1. There is no information on how the screening process was done by independent reviewers in methodology.

Response: Thank you for this comment, we added lacking information.

  1. The aim says ‘pediatric population’. Shouldn’t it be the pediatric ‘patient’ population?

Response: Thank you for this suggestion. The manuscript was checked by our native English-speaking colleague and ‘pediatric population’ was accepted as correct.

Round 2

Reviewer 1 Report

The manuscript is an interesting summarization of the current knowledge of galectin-3, which is of great value. I appreciate that the authors added more data from other observations and researches to enhance the impact of this manuscript. And other clinical biomarkers are talked about in the new manuscript. Overall, the manuscript and figures are logically organized and of good quality. 

Author Response

Thank you very much for this comment.

Reviewer 4 Report

I am not fully convinced with the author's response on improving the figures as mentioned in my previous review report. In the present form, the figures give no significant information (in fact the same information can be simply made in a sentence and the figure can be deleted). The authors should consider improving it. Thanks.

Author Response

Thank you for this suggestion. We prepared new version of the content as a figure. Hope it will meet your expectations.